

# Validation of a handheld β-hydroxybutyrate acid meter to identify hyperketonaemia in ewes

Carolina Akiko Sato Cabral Araújo[1], Antonio Humberto Hamad Minervino[2], Rejane Santos Sousa[3], Francisco Leonardo Costa Oliveira[3], Frederico Augusto Mazzocca Lopes Rodrigues[3], Clara Satsuki Mori[3] and Enrico Lippi Ortolani[3]

[1] Departamento de Medicina Veterinária, Universidade Federal Rural de Pernambuco, Recife, PE, Brazil
[2] Laboratório de Sanidade Animal, LARSANA, Universidade Federal do Oeste do Pará, UFOPA, Santarem, Brazil
[3] Departamento de Clínica Médica, Faculdade de Medicina Veterinária e Zootecnia, FMVZ, Universidade de São Paulo, São Paulo, SP, Brazil

Corresponding authors
Carolina Akiko Sato Cabral Araújo,
carolcbpr@gmail.com
Antonio Humberto
Hamad Minervino,
ah.minervino@gmail.com

## ABSTRACT

**Background:** The end of pregnancy is the period with the highest risk of occurrences of ketosis and pregnancy toxemia due to fat mobilization and increasing non-esterified fatty acids in the liver which are converted in ketone bodies, mainly β-hydroxybutyrate acid (BHB). This ketone body may also become elevated in the bloodstream. The present study validates the use of a handheld meter for determining the blood concentration of BHB and ascertaining the predictive value and accuracy of BHB measurements in diagnosing hyperketonaemia in ewes.
**Methods:** A total of 19, non-pregnant, crossbred ewes were subjected to 2 h of intravenous infusion of a saturated BHB solution. Over 6 h of evaluation, 247 blood samples were obtained in 13 sampling moments. The BHB concentration was measured by an enzymatic colorimetric method in an automated biochemical analyzer (gold-standard) and by a handheld meter using an electrochemical enzyme technique.
**Results:** There was a high correlation between both methods ($r = 0.98$; $P < 0.001$). Considering the blood BHB concentrations range 0.8–1.6 mmol/L for moderate ketosis the handheld meter presented sensitivity and specificity of 0.98 and 0.81, respectively. For severe ketosis (BHB ≥ 1.6 mmol/L) sensitivity and specificity were 0.99 and 0.75, respectively. Thus, the handheld device can be useful for diagnoses of cases of mild or severe pregnancy toxemia at field conditions.

## INTRODUCTION

The end of pregnancy and start of lactation are the periods of highest metabolic demand and consequently, the time of the highest risk of occurrences of ketosis and pregnancy toxemia (PT) in ewes (*Ortolani & Benesi, 1989*). Pregnancy toxemia is a metabolic disease caused by a negative energy balance that result in excessive lipid metabolism and ketosis

(*Araújo et al., 2018*). Occurs generally in ewes carrying two or more foetuses and has a high mortality rate (*Cal-Pereyra et al., 2015b*). In obese ewes, the fat storage formed around the rumen predisposes the occurrence of the disease by reducing dry matter intake (*Lacetera et al., 2001*; *Araújo et al., 2018*). Consequently, there is a great fat mobilization in an attempt to maintain the animal's energy supply. This mobilization increases initially non-esterified fatty acids in the liver that by its turn is partially transformed in ketone bodies, mainly β-hydroxybutyrate acid (BHB), but also acetoacetate and acetone (*Radostits et al., 2007*). The BHB may also become elevated in the bloodstream due to mobilization of the body's reserves, especially when there is a negative energy balance in lean pregnant ewes (*Radostits et al., 2007*; *Kalyesubula et al., 2019*). Blood ketosis can also evolve to a nervous condition (nervous ketosis) caused by a BHB-derived product, the isopropanol (*Araújo et al., 2014*).

So far, there is no agreement about the cut off values for BHB concentrations in sheep with severe PT. For severe status, some authors (*Scott et al., 1995*; *Balikci, Yildiz & Gurdogan, 2009*) consider more than 3.0 mmol/L, while others more than 1.6 mmol/L (*Lacetera et al., 2001*; *Kulcsár et al., 2006*) and a recent study from Uruguay pointed 2.26 mmol/L as indicative of moderate PT (*Cal-Pereyra et al., 2015a*). On the other hand, the authors agree that 0.8 mmol/L BHB should be cut point for moderate PT (*Kulcsár et al., 2006*; *Balikci, Yildiz & Gurdogan, 2009*).

Examining the blood BHB concentrations is fundamental for detecting hyperketonaemia in sheep and enables early diagnosis and a higher success rate in the treatment of PT (*Radostits et al., 2007*). Biochemical laboratory tests using commercial kits are considered to be the gold-standard (GS) for determining BHB concentrations, but semi-quantitative determinations of acetoacetate and acetone in urine by Rothera's test is frequently used for diagnosing ketosis. Nevertheless, BHB determination is preferred to Rothera's test because the former gives much less false-positives results and has better accuracy (*Kaneko, Harvey & Bruss, 2008*). Furthermore, BHB is more stable in blood, because it is not volatile and more abundant than the other two ketone bodies (*Kaneko, Harvey & Bruss, 2008*; *Benedet et al., 2020*).

A handheld meter (HHM) is available for investigating ketone bodies and glycemia in human blood (*Abbott Diabetes Care, LTD, 2007*). This equipment is widely accessible, has a low cost (less than one US dollar per analysis) and the advantage of being able to quantify BHB in blood samples. Recently, it has been validated for use in dogs and cats (*Hoenig, Dorfman & Koenig, 2008*), dairy cattle (*Voyvoda & Erdogan, 2010*) and sheep (*Panousis et al., 2012*; *Hornig et al., 2013*). Thus, this equipment can be easily used for a practical ewe-side diagnosis of PT (*Voyvoda & Erdogan, 2010*).

Previous studies that compared BHB concentrations in sheep venous whole blood measured by HHM and GS enzymatic method concluded that the BHB results did not differ between these methods (*Panousis et al., 2012*; *Hornig et al., 2013*). Nevertheless, both studies (*Panousis et al., 2012*; *Hornig et al., 2013*) analyzed BHB values from healthy sheep, with low blood BHB mean values and including limited number of samples with BHB concentration greater than 3 mmol/L, considered as the threshold for PT (*Scott et al., 1995*; *Balikci, Yildiz & Gurdogan, 2009*). Besides the HHM was validated for early detection of

**Table 1 Chemical composition of diet (hay and concentrate) used during the experiment.**

| Parameters | Coast-cross hay | Concentrate |
|---|---|---|
| Dry matter (%) | 84.1 | 87.0 |
| Crude protein (%) | 7.5 | 14.0 |
| Neutral detergent fiber (%) | 33.1 | 16.0 |
| Ether extract (%) | 1.9 | 2.0 |
| Ash (%) | 6.1 | 16.0 |

blood ketosis in sheep, the studies were performed with healthy animals and consequently with lower blood concentrations of BHB. Thus, we aimed to validate the HHM for the measurement of BHB in sheep with a wide range of blood BHB, including values corresponding to moderate and severe ketosis.

## MATERIALS AND METHODS

This research was approved by the Committee on the Ethics of Animal Experiments of the School of Veterinary Medicine and Animal Science, University of São Paulo (protocol 2142/2011), São Paulo, SP, Brazil.

### Animals

A total of 19 multiparous, non-pregnant, non-lactating Santa Inês crossbred ewes were used. Initially, sheep underwent a 30 days adaptation period, in which they were kept in collective pens. Sheep received a diet calculated as 2.7% of body weight (kg dry matter/d), which consisted of 50% coast-cross (*Cynodon dactylon*) hay and 50% commercial concentrate (Fri-Ovinos 22/70, Nutreco Nutrição Animal, Pitangueiras, SP, Brazil) (2.7%). Their mean ± standard deviation (SD) body weight was 50.9 ± 4.2 kg at the beginning of the study, after the adaptation period. Ewes were weighted weekly to correct the diet accordingly. The chemical composition of the diet is presented at Table 1. All animals had water and mineral mixture ad libitum.

### Study design

Sheep were subjected to a protocol for inducing hyperketonaemia adapted from the model described elsewhere (*Schlumbohm & Harmeyer, 2003*) to obtain samples with different concentrations of BHB. Before the BHB infusion, the animals were subjected to water and food fasting for 18 h. Then, each ewe received 5 mmol of sodium 3-hydroxybutyrate ($C_4H_7NaO_3$, ≥99.0%; Alfa Aesar®, Heysham, England) per kg of body weight in 360 mL fixed-volume solution with deionized water and pH adjusted to 7.4. Sodium 3-hydroxybutyrate varies from 191.5 to 335 mmol according to the ewe body weight. To ensure continuous infusion and better management of the animals during the induction, a plastic catheter was implanted in the right jugular vein (Intracat™; Becton Dickinson and Company, Franklin Lakes, NJ, USA). The protocol consisted of intravenous infusion of BHB solution initially at a rate of three mL per minute, being adjusted to ensure that the total BHB infusion time was 120 min for all animals.

During and after the infusion, blood was sampled through jugular venepuncture at baseline (T1) before the infusion and at 20, 40, 60, 80, 100 and 120 min (T2–T7) after the beginning of the induction. At the end of the infusion, six additional samplings were performed after 15, 30, 60, 120, 180 and 240 min (T8–T13). The rectal temperature was measured with a digital thermometer at the same times described above (Clinical Digital Thermometer TS-101; Techline, São Paulo, SP, Brazil). Animals were manually restrained for blood sampling. The induction model does not produce pain or discomfort to the animals. Complete clinical evaluations to assure animal welfare were done at each time-point, including all routine physical parameters and the ketone measurement in the urine using Combur-Test® (Roche Diagnostics, Basel, Switzerland). During the study the environment temperature varies from 22 to 25 °C.

## BHB measurements

In all the time-points two blood samples were obtained simultaneously. The first one was used for measuring BHB in the HHM (Optium Xceed®; Abbott Laboratories, São Paulo, SP, Brazil), using a disposable syringe of volume three mL that contained 0.1 mL of sodium heparin. This sample was homogenized and a drop (±1.5 µL of total blood) was immediately placed on the reactive strip indicated for the device, which gave the digital results in about 10 s (*Panousis et al., 2012*). The apparatus was calibrated according to the manufacturer's recommendations. The principle for measuring BHB in the HHM involves an enzyme-based electrochemical technique. Briefly, when the blood sample is applied to the β-ketone test strip, the blood BHB reacts with a chemical in the strip producing a small electrical current, which is measured and the sensor displays a result (*Abbott Diabetes Care, LTD, 2007*).

Another blood sample of four mL was collected into tubes with a vacuum system containing sodium fluoride and ethylenediaminetetraacetic acid as an anticoagulant (Vacutainer®, Becton Dickinson and Company, NJ, USA). These were homogenized by repeatedly completely inverting immediately after collection and were kept under refrigeration (4 °C to 6 °C) for a maximum of 2 h until they were processed. The tubes were centrifuged for 10 min at 697 *g* to separate the plasma, which was then stored at −20 °C until analysis by an enzymatic colorimetric method (*Williamson, Mellanby & Krebs, 1962*). Considering the induction protocol and the different collection times, 247 samples were obtained (19 ewes; 13 collection times) with different BHB concentrations, which were quantified by those two techniques.

## Statistical analysis

Data were analyzed throughout the Shapiro–Wilks normality test. Agreement between the handheld meter and the reference enzymatic colorimetric method were assessed using Deming regression, Passing–Bablok regression and Bland–Altman difference plot.
The later was used to determine the bias among the methods. Spearman rank correlation coefficient (*r*) was calculated.

Two threshold values for blood BHB were established as indicatives of moderate (0.8 to 1.6 mmol/L) and severe (≥1.6 mmol/L) ketosis (*Lacetera et al., 2001*; *Kulcsár et al., 2006*;

**Table 2 Passing–Bablok regression analysis comparing the handheld meter (HHM) and the gold-standard (GS) enzymatic colorimetric test.**

| Parameters | GS enzymatic colorimetric method (mmol/L) | Handheld meter (mmol/L) |
| --- | --- | --- |
| Lowest value | 0.11 | 0.10 |
| Highest value | 3.94 | 7.10 |
| Arithmetic mean | 1.25 | 1.90 |
| Median | 1.16 | 1.70 |
| Standard deviation | 0.89 | 1.56 |
| Standard error of the mean | 0.057 | 0.099 |
| Regression equation | HHM = −0.19 + 1.69 GS | |
| Systematic differences | | |
| Intercept A | −0.187 | |
| 95% CI | −0.215 to −0.162 | |
| Proportional differences | | |
| Slope B | 1.691 | |
| 95% CI | 1.657–1.724 | |
| Random differences | | |
| Residual standard deviation (RSD) | 0.137 | |
| ± 1.96 RSD interval | −0.268 to 0.268 | |
| Linear model validity | | |
| Cusum test for linearity | Significant deviation from linearity ($P < 0.01$) | |
| Spearman rank correlation coefficient | | |
| Correlation coefficient | 0.984 | |
| Significance level | $P < 0.0001$ | |
| 95% CI | 0.979–0.987 | |

**Note:**
   CI, confidence interval.

*Balikci, Yildiz & Gurdogan, 2009*). Using the two abovementioned cutoffs for moderate and severe ketosis we were able to convert the BHB concentration to positive/negative results (separately for both severe and moderate ketosis) and establish the sensitivity and specificity, positive predictive value, negative predictive value and accuracy of the handheld meter in relation to the gold-standard using classical epidemiological formulas (*Smith, 1995*). Statistical analysis was made with GraphPad Prism software (GraphPad Inc., La Jolla, CA, USA) considering 5% as significance level.

## RESULTS

Although the sheep were subjected to induction of hyperketonaemia, none presented clinical signs of PT, except intense ketonuria after 20 min of BHB infusion to the end of this process. Table 2 presents the Passing–Bablok regression analysis results.
The overall mean ± standard error (SE) value of BHB concentration using the GS was 1.25 ± 0.06 mmol/L, while the HHM presented a higher mean of 1.90 ± 0.01 (SE) mmol/L. The BHB concentration of the GS ranged from 0.11 to 3.94 mmol/L, while the results of the HHM ranged from 0.10 to 7.10 mmol/L. Figure 1 presents the Deming regression
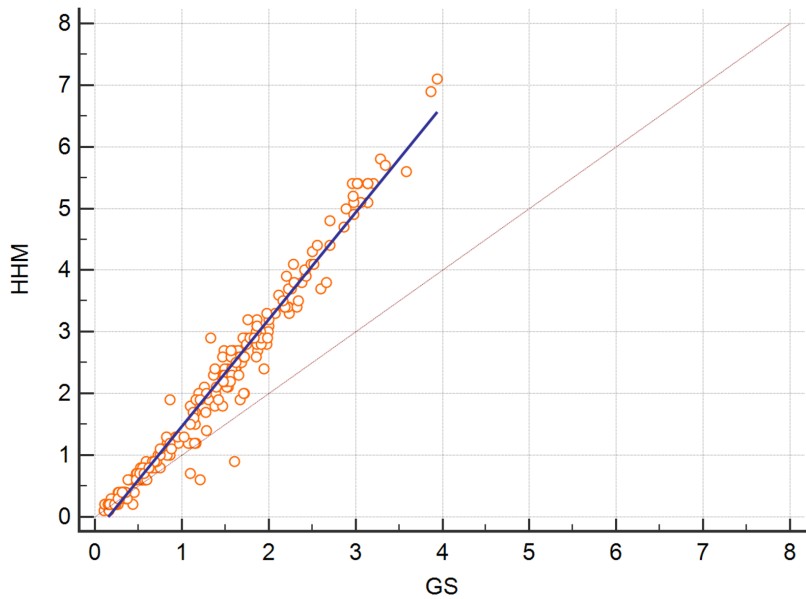

**Figure 1 Deming regression analysis between concentrations of β-hydroxybutyrate (BHB) blood measured by handheld meter (HHM) and gold-standard colorimetric method (GS) (n = 247).** The blue line indicates the regression line and the orange line denotes the identity line (x = y). Deming regression: HHM = −0.29 + 1.75 GS (n = 247). Pearson correlation coefficient r = 0.996.

analysis between concentrations of BHB measured by the two methods. The results from the HHM can be corrected according to the GS through the equation HHM = −0.19 + 1.69 × GS, according to the Passing–Bablok regression analysis (r = 0.984).

Figure 2 presents the Bland–Altman plot of the difference between BHB measured both methods (HHM minus GS) against the mean. The HHM results showed a bias of 0.65 ± 0.71 (SD) mmol/L (−0.74 to 2.04 95% limits of agreement). Figure 3 presents two Bland–Altman plots, the first one included samples with BHB values below 1.6 mmol/L, indicative of moderate ketosis but excluding the higher BHB values. The second plot considered only samples with BHB values below 0.8 mmol/L, indicative of physiological BHB values (*Lacetera et al., 2001*; *Balikci, Yildiz & Gurdogan, 2009*). When values were stratified the bias was reduced to 0.275 ± 0.38 (SD) mmol/L (−0.47 to 1.02 95% limits of agreement) for data excluding high BHB values and to 0.11 ± 0.21 (SD) mmol/L (−0.31 to 0.52 95% limits of agreement) for normal BHB range.

Table 3 evaluates the diagnostic result (positive or negative) from blood BHB concentration measured by HHM and GS method that will be indicative of moderate ketosis. Additionally, Cohen's kappa coefficient of agreement between tests is presented. For diagnostic of moderate ketosis, the HHM presented 19 false-positive and 3 false-negative results, whereas for severe ketosis diagnose, 41 false-positive and 1 false-negative, in comparison to GS. It is noteworthy that the ketonemia induction model resulted in 52 samples (21%) with BHB concentration above 3.0 mmol/L.

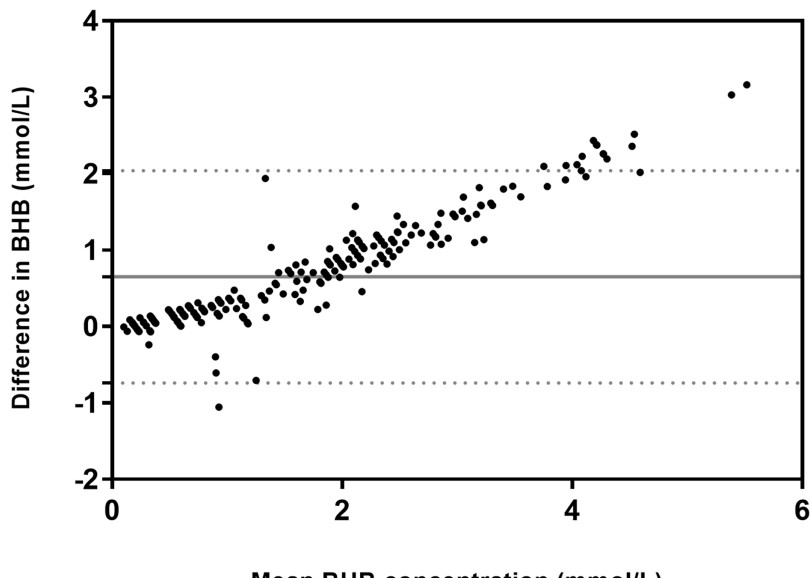

**Figure 2 Bland–Altman plot of the difference between BHB measured by the handheld electrochemical meter and the BHB measured using the gold-standard colorimetric method against the mean BHB for both methods.** The solid horizontal line is the mean bias (0.65 mmol/L) and the two horizontal dashed lines represent the 95% CI for agreement.

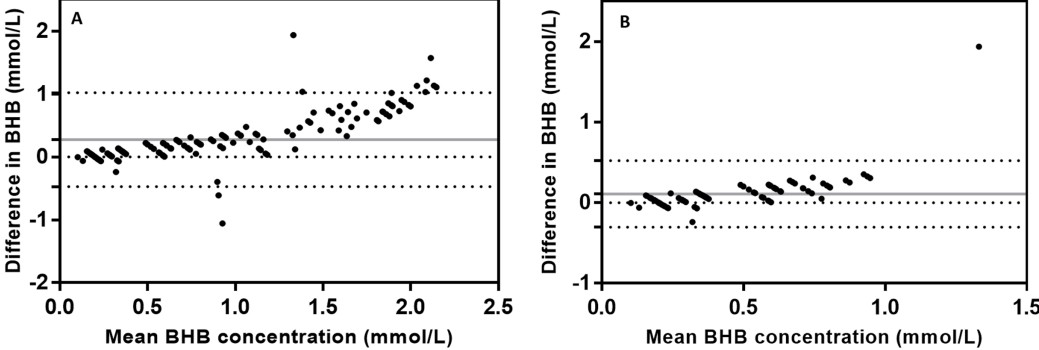

**Figure 3** (A) Bland–Altman plot considering only BHB values below 1.6 mmol/L ($n = 162$) of the difference between β-hydroxybutyrate (BHB) measured by the handheld electrochemical meter and the BHB measured using the gold-standard colorimetric method against the mean BHB for both methods. The solid horizontal line is the mean bias (0.27 mmol/L) and the two horizontal dashed lines represent the 95% Confidence Interval (CI) for agreement. (B) Bland–Altman plot considering only BHB values below 0.8 mmol/L ($n = 101$) of the difference between BHB measured by the handheld electrochemical meter and the BHB measured using the gold-standard colorimetric method against the mean BHB for both methods. The solid horizontal line is the mean bias (0.11 mmol/L) and the two horizontal dashed lines represent the 95% CI for agreement.

The rectal temperature increased (Fig. 4) as the BHB infusion started and kept higher as compared to the pre-infusion time. The mean rectal temperature at the selected time points where we observed false-positive results for the severe ketosis ($n = 41$; 38.9 ± 0.6 °C) were higher when compared to the baseline ($n = 24$; 38.5 ± 0.8 °C) ($P < 0.05$).

**Table 3 Diagnostic results (positive or negative) from blood β-hydroxybutyrate (BHB) concentration measured by handheld meter and gold-standard colorimetric method indicative of moderate ketosis (BHB 0.8–1.6 mmol/L) or for severe ketosis (BHB ≥ 1.6 mmol/L).**

| | | Moderate ketosis (BHB 0.8–1.6 mmol/L) | | | Severe ketosis (BHB ≥ 1.6 mmol/L) | | |
|---|---|---|---|---|---|---|---|
| Methods | | Gold-standard | | | Gold-standard | | |
| | | Positive | Negative | Total | Positive | Negative | Total |
| Handheld meter | Positive | 143 | 19 | 162 | 84 | 41 | 125 |
| | Negative | 3 | 82 | 85 | 1 | 121 | 122 |
| | Total | 146 | 101 | 247 | 85 | 162 | 247 |
| Sensitivity | | 0.98 | | | 0.99 | | |
| (95% confidence interval) | | [0.94–1.00] | | | [0.94–1.00] | | |
| Specificity | | 0.81 | | | 0.75 | | |
| (95% confidence interval) | | [0.72–0.88] | | | [0.67–0.81] | | |
| Accuracy (95% confidence interval) | | 0.91 [0.87–0.94] | | | 0.823 [0.78–0.88] | | |
| Cohen's kappa coefficient | | 0.811 | | | 0.661 | | |

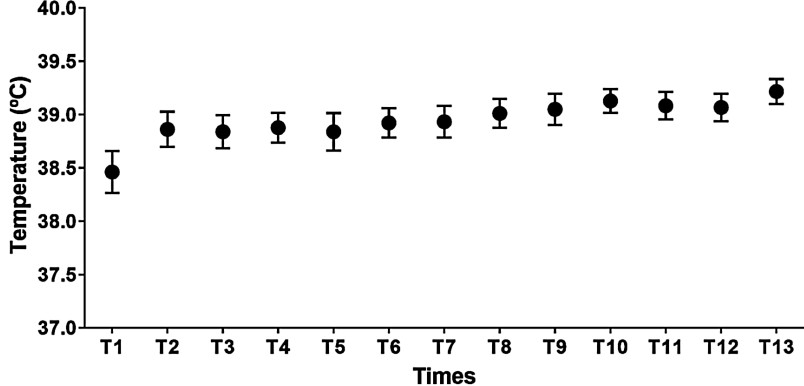

**Figure 4 Mean values and standard error of rectal temperature (°C) during the experiment.** β-hydroxybutyrate infusion stared at T2 and finished at T7.

## DISCUSSION

The infusion of BHB solution caused a rapid increase in the BHB blood concentration values that were sustainably high throughout the infusion period and decreasing continuously thereafter. Nevertheless, as the blood concentration surpasses 1.0 mmol/L BHB, the renal threshold, an intense ketonuria collaborate to decrease or to avoid an extreme hyperketonaemia in the blood.

The current model of induction of hyperketonaemia produced greater number of samples (n = 52; 21%) with blood BHB values higher than 3.0 mmol/L measured by HHM than those obtained by other authors: 18 out 193 (9.32%) (*Panousis et al., 2012*) and 8 out 465 (1.72%) (*Hornig et al., 2013*), being the later, the study that validated the HHM for

sheep. The presence of more samples with higher blood BHB concentration turn the validation of the HHM more reliable than those carried out in other early studies.

Although there are previous literature regarding the use of handheld meter in dairy cow, they were performed with healthy animals (*Nielen et al., 1994*; *Carrier et al., 2004*; *Kupczyński & Cupok, 2007*), thus, resulting in limited data of affected animals (i.e., animals with moderate or severe ketosis). In sheep, similar studies evaluated healthy animals (*Firat & Özpinar, 2002*; *Panousis et al., 2012*) and one study used feed restriction in late gestation to increase blood BHB concentration (*Cal-Pereyra et al., 2015a*). No previous studies used our BHB-infusion methodology, which resulted in a higher number of animals with increased blood BHB and therefore a more robust dataset for sensitivity and specificity analysis. Additionally, our study is the first report using Brazilian beef breed (Santa-Inês) at tropical conditions.

Overall mean BHB concentration was higher in the HHM than the GS ($P < 0.05$). These results were different from those recorded by previous reports (*Panousis et al., 2012*; *Hornig et al., 2013*) that described a similar or greater mean values for BHB measured by GS than HHM. Previous studies with cattle show that the difference between HHM and GS enlarged as the BHB levels were higher than 3.0 mmol/L (*Megahed et al., 2017*). As in the current experiment, a reasonable amount of the blood samples (21%) surpass 3.0 mmol/L of BHB, this could increase the difference of the overall means obtained by HHM and GS. This difference could interfere with the results since there was a strong positive correlation ($r = 0.99$) and a high sensibility (0.99), negative predictive values (0.99) and accuracy (0.87).

Using the HHM for the diagnostic of BHB concentration indicative of moderate ketosis (0.8–1.6 mmol/L) when compared with GS, we found a high sensitivity (0.98) and specificity (0.81) and a perfect agreement at Kappa test, similarly as previous reports with healthy animals (*Dawson, Carson & Kilpatrick, 1999*; *Voyvoda & Erdogan, 2010*; *Panousis et al., 2012*; *Pineda & Cardoso, 2015*; *Macmillan et al., 2017*). Using the HHM for the diagnosis of blood BHB concentration indicative of severe ketosis (BHB ≥ 1.6 mmol/L), only one false-negative case was detected indicating that HHM has a very high sensitivity (0.99). On the other hand, we found 41 false-positive results suggesting that decreased somehow the specificity (0.75). The methods had a substantial agreement at Kappa test. Conversely, previous reports (*Panousis et al., 2012*; *Hornig et al., 2013*) using the same equipment obtained very high specificity.

According to *Iwersen et al. (2013)* and *Megahed et al. (2017)* the higher the blood temperature, the greater is the BHB concentration measure by HHM. The rectal temperature increased as the BHB infusion started and kept high as compared to the pre-infusion time. Most of the false-positive results occurred within the first 60 min of infusion and between 15 and 60 min post-infusion when the BHB blood concentration was increasing and decreasing, respectively. By these times the rectal temperature increased as well, probably by a small quantity of pyrogen in the solution that caused a short and slight hyperthermia. According to *Bligh (1957)*, there is a very high positive correlation between rectal and blood temperature. Thus, in the early ascending and descending curve when the BHB blood levels were between 1.0 and 1.5 mmol/L, measured by GS, most of

the false-positive samples had values superior to 1.6 mmol/L probably caused by higher blood temperature.

## CONCLUSION

An overview of the results within the different ranges of β-hydroxybutyrate acid permits to affirm that handheld meter is sufficiently accurate and sensible to detect hyperketonaemia in sheep, which recommends its use to provide reliable, rapid, ewe-side early diagnosis of pregnancy toxemia in sheep. This was this first study to validate this equipment in tropical conditions.

### Ethics statement

This research was approved by the Committee on the Ethics of Animal Experiments of the School of Veterinary Medicine and Animal Science, University of São Paulo. All animals were treated with high standard (best practice) of veterinary care and with animal welfare.

## ACKNOWLEDGEMENTS

The authors are grateful to the anonymous reviewer (reviewer #3) for his/her commitment to the peer reviewer process providing four rounds of revisions that resulted in a great improvement of our manuscript.

### Funding

This research was funded by the Brazilian National Council for Scientific and Technological Development (CNPq) and Fundação de Amparo à Pesquisa do Estado de São Paulo (FAPESP). Antonio Minervino was a recipient of a research productivity fellowship from CNPq. The funders had no role in study design, data collection and analysis, decision to publish, or preparation of the manuscript.

### Grant Disclosures

The following grant information was disclosed by the authors:
Brazilian National Council for Scientific and Technological Development (CNPq).
Fundação de Amparo à Pesquisa do Estado de São Paulo (FAPESP).

### Competing Interests

The authors declare that they have no competing interests.

### Author Contributions

- Carolina Akiko Sato Cabral Araújo conceived and designed the experiments, performed the experiments, analyzed the data, prepared figures and/or tables, authored or reviewed drafts of the paper, and approved the final draft.
- Antonio Humberto Hamad Minervino conceived and designed the experiments, performed the experiments, analyzed the data, prepared figures and/or tables, authored or reviewed drafts of the paper, and approved the final draft.

- Rejane Santos Sousa conceived and designed the experiments, performed the experiments, prepared figures and/or tables, and approved the final draft.
- Francisco Leonardo Costa Oliveira performed the experiments, prepared figures and/or tables, and approved the final draft.
- Frederico Augusto Mazzocca Lopes Rodrigues performed the experiments, authored or reviewed drafts of the paper, and approved the final draft.
- Clara Satsuki Mori performed the experiments, authored or reviewed drafts of the paper, biochemical analysis, and approved the final draft.
- Enrico Lippi Ortolani conceived and designed the experiments, analyzed the data, authored or reviewed drafts of the paper, and approved the final draft.

## Animal Ethics

The following information was supplied relating to ethical approvals (i.e., approving body and any reference numbers):

This research was approved by the Committee on the Ethics of Animal Experiments of the School of Veterinary Medicine and Animal Science, University of Sao Paulo (Protocol #2142/2011).

## Data Availability

Raw data are available in the Supplemental Files.

## Supplemental Information

Supplemental information for this article can be found online at http://dx.doi.org/10.7717/peerj.8933#supplemental-information.

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
