# Peer review of "Validation of a handheld β-hydroxybutyrate acid meter to identify hyperketonaemia in ewes"

_PeerJ, doi:10.7717/peerj.8933_

## Round 0.1 · original submission · Major Revisions

Dear authors, several issues pointed out by reviewers must be addressed before further consideration of your manuscript.

Reviewer 1 ·

Basic reporting

In my opinion, the research work can be accepted, but some issues need to be taken into account.
The work really, although interesting, does not contribute much. It is a further investigation on the use of portable meters to detect BOHB values and there are multiple publications on this aspect. Some of them comparing different meters, others comparing with more complex (and more expensive) laboratory techniques, others made in different animal species such as cattle, goats and sheep, and even comparing 2 and 3 portable devices (among them) existing in the market, to check its sensitivity and accuracy.

Really the only novelty is the method of induction of ketonemia, even causing severe ketonemias. In most of the publications, animals that present spontaneous disease are sampled, or induce ketonemia (or pregnancy toxemia) by combining food restriction, fasting and advanced gestation.
It is true that most publications have fewer samples, but in my opinion it is justified by the previous reason. And when cattle are used, the number of samples is usually even much wider. In my opinion, the sensitivity of the portable meter is indifferent that it is made for one species than for another.
It seems that there is interest in not mentioning the name of the meter tested (Optium Xceed®). I think it should be indicated at least in the Abstract, in the Discussion and in the Conclusions.

Experimental design

..

Validity of the findings

..

Additional comments

Specific comments

Abstract
Include the name of the hand meter tested (Optium Xceed®) in the Abstract.

Introduction
Βeta-hydroxybutyrate acid has not been correlated with the abbreviation BHB the first time it is used in the text (line 58). Yes it is done in the Abstrac, but I would recommend doing it in both places
Unify abbreviations and always use them. HM (as handheld meter) is used, but HH and HHM are also used in figure 1. SG (gold standard) and enzymatic colorimetic method (line 362, title Table 1, and within the same table 1). mM / L and mmol / L: unify
Remove capital letters that are not strictly necessary (Less, line 80; Handheld, line 362; Enzimatic line 362, etc)

Materials and Methods
How was the BHB cut point set? In addition to those indicated, I would recommend revising the reference Cal et al, 2015. In this publication, which is also taken as a reference for the diagnosis of subclinical pregnancy toxemia, the value of 2.26 mmol / l is set.
Clinical and Subclinical are continually referred to, and become synonymous with Severe and Moderate ketosis. It is an important mistake. The clinical and subclinical concept refers to the presence of visible or easily detectable clinical signs, not the more or less high BHB value. In Material and Methods there is no reference to this classification in clinical and subclinical.
Neither in Material and Methods is it indicated if other clinical signs are collected, in addition to the rectal temperature. Therefore it is difficult to classify the disease the clinical and subclinical.
Even on line 171 it is indicated: “… none presented any clinical sign of PT”. How can we talk about sheep with clinical signs in the rest of the manuscript, if it has been said that none presented clinical signs? Check.
In Material and Methods it is not indicated that ketoneuria is measured, but then it is indicated “… an intense ketonuria after 20 minutes” (line 172). Has ketoneuria really been measured? Check.
Line 145: “… containing sodium fluoride anticoagulant (Vacutainer®, ..)”. Sodium fluoride is not an anticoagulant, but in these vacuum tubes it is usually combined with potassium oxalate or disodium EDTA, which do act as anticoagulants. Sodium fluoride stabilizes glucose levels, preventing glycolysis.
Line 167: "... with GraphPad Prism software (GraphPad Inc., La Jola, CA, USA) ..." change to "... with GraphPad Prism software (GraphPad Inc., La Jolla, CA, USA)"
I would recommend making a specific section, or at least an individual paragraph, about the statistical study. How has specificity and sensitivity been assessed?

Figures and Tables
Review all titles in Tables and Figures. Take into account the Clinical and Subclinical indications, and possibly change for moderate and servere ketonemia, and Handheld meter (HM) and gold-standard enzymatic colorimetic test (GS).
I do not understand the titles in Table 3. Subclinical ketosis should be changed to BHB ≥ 0.8 mmol / l and BHB ≥ 1.6 mmol / l. Perhaps I would recommend removing the standard gold line and indicating it in the title of the figure.
Figure 3. Bellow is incorrect. Replace with below. (Lines 395 and 400)

Discussion
In general the discussion is correct, but I would recommend reviewing the references below and including them in the discussion.
Line 243 and following: The rectal temperature is increased, probably by a small quantity of pyrogen in the solution ... In my opinion the stress of manipulation, sampling and immobilization, may be sufficient to induce a slight elevation of the body temperature. And that would justify the slight hyperthermia more than the possible presence of pyrogens.

References

I would recommend reviewing the following references, and perhaps including them in the discussion of this manuscript

Cal-Pereyra L, Benech A, González-Montaña JR, Acosta-Dibarrat J, Da Silva S, Martín A. 2015. Changes in the metabolic profile of pregnant ewes to an acute feed restriction in late gestation. New Zealand Veterinary Journal, 63:3, 141-146;
Carrier J, Stewart S, Godden S, Fetrow J, Rapnicki P. Evaluation and use of three cowside tests for detection of subclinical ketosis in early postpartum cows. J DairySci 2004;87(11):3725-3735.
Firat A, Ozpinar A. 2002. Metabolic profile of pre-pregnancy, pregnancy and early lactation in multiple lambing Sakiz ewes 1. Changes in plasma glucose, 3-hydroxybutyrate and cortisol levels. Annals of Nutrition and Metabolism 46, 57–61.
Kupczynski R, Cupok A. Sensitivity and specificity of various tests determining beta-hydroxybutyrate acid in diagnosis of ketosis in cows. Electronic Journal of Polish Agricultural Universities Series Veterinary Medicine 2007;3(10):1-15.
Mair, B., Drillich, M., Klein-Jöbstl, D., Kanz, P., Borchardt, S., Meyer, L., ... & Iwersen, M. (2016). Glucose concentration in capillary blood of dairy cows obtained by a minimally invasive lancet technique and determined with three different hand-held devices. BMC veterinary research, 12(1), 34.
Nielen M, Aarts MG, Jonkers AG, Wensing T, Schukken YH. Evaluation of two cowside tests for the detection of subclinical ketosis in dairy cows. Canadian Vet J 1994;35(4):229.

Reviewer 2 ·

Basic reporting

This research paper presents the test of a handheld “ewe-side” ketometer device. There are already published relevant data concerning the same measurements in ruminants, but this research offers an additional approach to a known subject. The innovative approach of the paper is that the ewes were administered beta-hydroxybutyrate acid (BHB) intravenously and the artificial hyperketonemia enables the researchers to test for high blood BHB values, not easily found in clinical practice and natural occurring cases. Study design is good and the idea of BHB infusion and measurements over time is interesting. Study was performed well, but the results of the study are not presented well, on the contrary, the results are not clear and reader cannot actually understand the paper. Significant corrections and changes are suggested in the presentation and the selection of data that are the results of the study. In the form that are presented now, results and data do not comply with the study design, which is promising and well performed, and cannot justify publication of this work.

English language is adequately used, although some mistakes could be corrected L54, L57, L86, L103, L127,…
Literature is adequate in selection, use and presentation and format.
Tables are not presetned well, are confusing and do not help the presentation of results.
Data selection and presetnation is not adequate.
Detailed comments:
Abstract: Wrong numbers of ewes and samples are written (comment below).
L81, though the most common is US dollar, it would be better to define it in the text as US dollar (USD)
L101, it is mentioned that 19 multiparous ewes, but in abstract (L27-28) it is claimed that 24 ewes were used which is not correct. In the same way the total samples were 13X19=247 (L152-153 and as seen in Table 3) and not 312 (13X24) as claimed in L29

L128. Physical evaluation could be described, as it is important either for the welfare of animals and for the interesting clinical point of artificial hyperketonemia induction. For example the ketonuria mentioned later on, how was measured? Were there changes in heart/respiratory rate? Preliminary nervous signs? Any clinical changes in a ewe with BHB >3 mM/L would be interesting to be mentioned. It is also useful to mention the climate condition, especially temperature during the sampling time.
L139-L143 This part does not offer anything significant, it can be omitted

L362 replace Table 1 with Table 2
The results as described in test and presented in Tables are not comprehensive and actually confusing.
L174-177 these results are not presented in table 1, difficult to understand where are obtained from. For example: The overall mean value of BHB concentration using the GS was 1.85±1.45 mM/L, (according to Table 1 is 1.90±1.56!!) while the HM presented a higher mean of 3.10 ± 2.90 mM/L (P <0.05) (according to table 1 is the opposite, HM is lower in mean 1.25 ± 0.89 mM/L!). The BHB concentration of the GS ranged from 0.10 to 5.65 mM/L (according to table 1 ranged from 0.10 to 7.10 mM/L!), while the results of the HM ranged from 0.10 to 10.90 mM/L. (according to table 1 ranged from 0.1050 to 3.9360 mM/L!)
In other words, according to Table 1, GS method obtains significant higher values, which is in contrast to the results presented in Table 2. Moreover, in this part of the text results of both tables (1,2) are presented, and in making confusion bigger table 2 is named table 1 in L 362

L182-184 and Table2
In Table 2, authors allocate the mean values obtained according to BHB range in 3 categories (0-79 mΜ/L)/ 0.8-1.6 mΜ/L and >1.6 mΜ/L and then present the mean values obtained from HM and GS measurements. The results present the big difference between the mean values obtained from the 2 different methods of BHB measurement. But, in case the difference of mean values is so wide e.g.1.82 vs 1.27 for the category (0.8-1.6 mΜ/L) is questionable how is possible to have the same number N=61 of values in both measurement methods. If Handmeter (HM) gives constantly higher results, than GS, it is curious to have the same number N=61 of measurements for both method in the same wide range (0.8-1.6 mΜ/L) category. It could be assumed that due to wider range of valued obtained from HM method, the numbers of results in each category would differ between the methods. Anyway, this Table (table 20 is more confusing than helping for the reader, only the overall –line results make sense.
It would be more informative to compare in the same sample the different values obtained from HM and GS. Also it would be more adequate to compare mean values of each sampling time T1-T13 and present the differences in values measured with HM and GS method. It would be also more interesting from clinical point of view, as to see how infusion and excretion of BHB in blood changes. Authors designed a trails with BHB IV infusion and repeated measurements on certain times frames and do not present these results! On the contrary they mix mean results of both methods (table 2) offering confusion to the reader.

Again, in table 3 the way columns and line are allocated is impossible for the reader to understand which are positive and negative by which method, so as to understand false and true positive ones.

L205-207: These results are not presented and are actually a basic finding according to the design of the study.
L208, L172: Ketonuria is mentioned here, but it was not mentioned in the material and methods. Urine BHB was measured to all animals in each sampling time? It could be added in the relevant section
L210-211: These results are not presented anywhere in the study
L233: sensibility?
L241-243 and L247-249. As mentioned above these results are not presented anywhere in the paper and are actually important and one of the innovation of this study, but authors simply omitted form results and add suddenly this information in the discussion part.

Paper is self-contained.

Experimental design

Experimental design is adequate for the well defiend reseasrch quastion. Presentation of the design in the material and methods section needs slight improovement, but in general are described well.
Statistical analysis is adequate and accurate.
Adequate and relevant to the aims of the study, methods followed, technical and ethical standards are fulfiiled.

Validity of the findings

The major problem of the submitted paper is the selection of data that authors presented and the way these data are presented.
Desing of the study enables repated measurement before during and after intravenous infusion of BHB in ewes. According to the study desing, research could harvest valuable measurements of variable blood BhB concentrations including high and very high values and their change in certain time frames. Unfortunately authors do not present these results, that could be the nevel thing of thiw research. On the contrary present insuffuciently a mixture of mean values, with no time reference in a very confusing and disrupted way.
This makes thiw work incomprehensive and actually hinters the soundness of the findings/
Under this point of view discussion looks poor and also the conclusions.

Additional comments

A nice idea and well designed study is actually wrong by the selected results The data selected for presetnation and the way that are presented, make it very diffcult to understand the findings of the study.
More details ans suggestion are writen in the basic reporting section of the review.

Reviewer 3 ·

Basic reporting

-In general, the syntax is not clear. It is difficult to follow the text. The structure of the sentences should be improved. For example, keep sentences shorter (about two lines).
-Please, check layout for submitting to this journal
-Lines 1 to 20: Please check the journal layout to identify affiliation, how to report an affiliation, and how to indicate and the corresponding authors and the information to be included. (See template of the journal: https://peerj.com/about/author-instructions/
-Line3 to 5: Use numbers instead of letters for the affiliation of the authors.
-Not sure that the Short title is needed for that journal (Line 21)
-Check layout required for the abstract: https://peerj.com/about/author-instructions/
Subheadings must be bold, followed by a period, and start a new paragraph e.g.
Background. The background section text goes here...
-Line 24: Please add information regarding the Background of the question, instead of starting with the aim of the paper.
-Line 27: Be consistent with the abbreviation selected to use. BHB or BHBA? Or define both if needed.
-Line 33-36: Sentence not clear
-Line 33: Please, report the p-value in the format required by the journal. If not clear, in the journal guidelines, please keep a space at both sides of the “<”.
-Line 37: What do you mean “in practical conditions”?
-Line 38-39: Start each Keyword with capital letters. Check the layout required by the journal.
-Line 41: This journal does not require “Implications”. Please check the sections required. However, it cannot be a unique sentence and it is too long.
-Line 51: Do not start a sentence with an abbreviation.
-Line 62-63: The reference from Benedet et al. 2019 is not correctly used because it is a review in cows, not in sheep or in ruminants in general. Please check that the references in the text are correctly used and placed.
-Line 78-80: Include a references for that sentence.
-Line 80: Do not use the capital letters in a sentence between brackets.
-Line 82-84: Check the structure of the sentence.
-Line 98: Make subsections to make it easier to follow. For example: animals, sampling collection, analysis, statistical analysis, etc.
-Line 101: Specify the country.
-Line 103: Revise the sentence.
-Line 159: Indentation of the paragraph.
-Line 167: Check how to report states and countries and keep it through all the test.
-Line 200: the term “somehow” is colloquial, be more specific.
-Line 172: If none of the animals presented any clinical sign of PT, the study does not respond correctly to the objective presented in Line 95.
-Some references are very old.

Experimental design

-Line 24: Abstract information does not match the material and methods. In the abstract, it is indicated 24 ewes, but in the material and methods, they are indicated 19. The number of samples in the abstract also is for 24 ewes, not for 19. Moreover, the Approved Ethical Committee is for 20 ewes. Also, the blood samples indicated in the abstract (312) are consistent with 24 ewes, not 19.
-Line 102: Indicate also the body condition score.
-Line 102: It was measured at the beginning of the adaptation period? Please, specify.
-Line 105: Provide chemical composition of the hay and the composition and chemical composition of the concentrate.
-Line 111: Check sentence. The use of “elsewhere” is not adequate for that sentence. Better refer directly to the reference.
-Line 111-112: Check sentence to make it easier to follow.
-Line 112: “Then, each ewe received”
-Line 112-115: Check punctuation to make it easier to follow.
-Line 111-119: Not clear the steps followed. Reconsider the order of the sentences.
-Line 123: Where does also in the left external jugular.
-Line 123-124: Not clear. Are those sampling times reported as minutes after the infusion?
-Line 126: What means SP? It also indicates the country.
-Line 150: Check sentence. The use of “elsewhere” is not adequate for that sentence. Better refer directly to the reference.
-Line 152-153: Number of samples and ewes do not match with the ones in the abstract.
-Line 172: Ketonuria refers to urine. Was urine check for that? How?

Validity of the findings

-Table 1: Explain all abbreviations
-Error in the number of the Tables. Please check it.
-The second “Table 2”, delete the comma after “et al.,”.
-Figure 1. Abbreviation of the y-axis is not defined in the caption of the figure. Also indicate the number of samples.
-Figure 4: x-axis indicates that is “sampling time”. Indicate when the infusion started and finished.
-Line 199-200: Is the temperature pre-infusion significantly different from the temperature infusion and post-infusion?
-Line 200 to 202: Sentence not clear.
-Line 201: Which times were classified at false-positive samples?
-Line 205-207: Sentence not clear.
-Line 208: How was ketonuria evaluated? It is not described in the material and methods section.
-A graph with the BHB concentration at each sampling point for both methods could be helpful to follow the paper.
-Line 210: Why? Which is the differences between the studies?
-Line 216: Any explication for that?
-Line 230: This is a result sentence not a discussion.
-Conclusions are poor.

---

## Round 0.2 · Minor Revisions

As you can see in the reviewers' comments letter, some points must be addressed yet before acceptance of your article. We look forward to hearing from you.

Reviewer 1 ·

Basic reporting

Peer J; 43743v1 VALIDATION OF A HANDHELD Β-HYDROXYBUTYRATE ACID METER TO IDENTIFY HYPERKETONAEMIA IN EWES

The manuscript has improved substantially with the revision. Practically the authors have taken into account the indications of all the reviewers and made the appropriate modifications, and in my opinion it is ready to be published.
I would only recommend some small changes, which would improve the article, some referring to the figures and tables and other grammatical errors.

Experimental design

..

Validity of the findings

..

Additional comments

Line 28: "... Βeta-hydroxybutyrate acid (BHB), ..." replace with "... beta-hydroxybutyrate acid" or "... β-hydroxybutyrate acid (BHB) ...". Do not capitalize B. Use this same criterion on line 59 and whenever this word is used.
Line 175. Animals, italicize. Unify with other subtitles
Line 114: remove “Chemical composition of the diet”
The word “bellow” continues to appear in the title of Figure 3. It is an error that has not been corrected. Replace with below.
After the modifications made to the tables and figures, it is still difficult to understand them. Especially review table 2, try to make them easier to understand. I propose the following Table 2.
Classification in moderate (BHB: 0.8-1.6 mmol / L) and severe (BHB ≥ 1.6 mmol / L) ketosis. Take into account the entire manuscript. Especially check line 282 (should put BHB ≥ 1.6 mmol/L), Table 2, Figure 2, etc.

Annotated reviews are not available for download in order to protect the identity of reviewers who chose to remain anonymous.

Reviewer 2 ·

Basic reporting

This is the second revision of the manuscript.
Authors, paid attention and made special effort to answer all comments and questions and made significant corrections and changes in the manuscript. In the form it is now it is much more comprehensive and to the point. In general, the revised manuscript presents clearly the work and stresses the results and the novelty of the study. As it was stated and in the first revision, a direct comparison of BHB values measured by 2 methods and obtained each sampling time, would be of interest, since the induced hyperketonaemia is the novel thing of the study. Authors claimed this was published elsewhere, which also diminishes the value of the manuscript. I didn’t notice any changes in tables and figures in the revised submission, as they are exactly as submitted for first time. Given that it is still not possible to read the Table 2, which is positive or negative by which method. It is not clear in the way presented.

Experimental design

Experimental design is as at the first submission, but authors made all the recommended changed in the presentation of the design and the material and methods section, so as to be clear and comprehensive. Experimental design is simple, clear and adequate for the aims of the study, so are the laboratory techniques and the statistical methods performed.

Validity of the findings

Discussion is improved and remains relevant, supports the findings and enlightens the results of the study. Conclusions are adequate and literature is up to date and representative of the study.

Additional comments

The re-submitted revised version of the manuscript offers significant changes and improvements, following reviewer’s suggestions and comments. In general, manuscript is better, sufficient and adequate considering its aims, its methods and results. There are some minor comments, that would meke the manuscript better:
(Line numbers refer to tracked changes word archive)
L26-28. Language needs improvement as to be more comprehensive. A suggestion:
The end of pregnancy is the period with the highest risk of occurrences of ketosis and pregnancy toxaemia due to fat mobilization and increasing non-esterified fatty acids in the liver which are converted in ketone bodies

L125 “Chemical composition of the diet” This is not presented or discussed.

It is still not possible to read the Table 2, which is positive or negative by which method.It is not clear to the reader, the positive and negative results should be presented clearly for both methods used.

L232-234 It is noteworthy that the ketonemia induction model resulted in 61 points of data (24.7%) with BHB concentration above 3.0 mmol/L.
Difficult to understand what is noteworthy. According to raw data se provided In the raw there are 63 points –values higher than 3,0 mmol/lt, but almost all (52) of them are referred to HHM. This could be stress also in discussion, as enlighten that HHM measure and give result in high BHB values, but there are usually false positive! according to GS method

L210 where instead of were.

Reviewer 3 ·

Basic reporting

- Some words seem to be in grey (for example L28 and Lines 30, or Line 60).
- Check punctuation used (see Line 28).
- Sentence too long (Line 26-28).
- Line 28: “is” or “are”?
- Line 30: indicate the brand.
- Line 30: Instead of indicating here the brand of the handheld meter, I think it is better to indicate it on the Methods (Line 37).
- Line 40: specify “respectively” also for 0.98 and 0.89.
- Revise punctuation and sentences too longs. That makes difficult to follow the paper. For example, Line 39-42, or Line 55-56.
- Check English grammar (Line 43: revise “at the field conditions”
- Line 51: Rewrite as suggested “and, consequently, of the highest risk of occurrences…”
- Line 53: What do you mean with “conditioned”? Sentence not clear.
- Line 56: Not clear. The disease reduces the DM intake, or predisposes the disease because of the decrease of DM intake?
- Line 61: To which ketone body is referring?
- Line 71: Revise “Monitoring of BHB concentrations”
- Line 78: “is most” or “is more”?
- Line 78-79: “in blood”
- Line 79: Revise the sentence. Suggestion: “it is not volatile and more abundant than the other two ketone bodies”
- Line 81-83: Sentence not clear. Check the correct place to indicate the name of the handheld meter used, and to what is referring “this”.
- Line 85: Confusing structure of the sentence. Revise “It has recently also been validated for use”
- Check extra spaces between words, such the one in Line 85 (for use in dogs”).
- Line 92-95: Sentence difficult to follow.
- Line 95-97: Sentence difficult to follow. Not clear “with lower blood concentrations of BHB”. Maybe it is easier to indicate the range where it has been validated.
- Line 98: Aim not clear. “wide” or “wider”? What means “from predictive values to ewes with clinical signs of PT”?

Experimental design

- Line 104: Although I imagine that the University of Sao Paulo is in Sao Paulo (Brasil), you think that you need to indicate the city (or state) and the country.
- Line 108: Revise sentence “a 30 days adaptation period of 30 days”.
- Line 111-Line 112: Sentence not clear.
- Line 114: Not clear: What do you mean “to correct the diet”?
- Line 113: I’m not familiar with this specific breed. Is the average BW reported in agreement with that breed? Or they presented under or over BW?
- Line 115: Sentence not finished.
- Line 115: The chemical composition of the coast-cross grass and the commercial concentrate is relevant for the readers. So they can determinate by themselves if the animals were correctly fed. I strongly recommend adding as much as you can regarding that point. At least, for the commercial concentrate, it should be easy to ask the company that produced it to provide that information. About the hay, I do not know if you produced it, or you buy it for the experiment, but usually, the chemical composition is determined to see if animals’ requirements are fulfilled.
- Line 118-125: Did the animals had free access to mineral salt block and water?
- Line 119-120: Check sentence order and the use of a comma.
- Line 120: “&” or “and”?
- Line 124: administered?
- Line 128-129: Sentence not clear: “with a total of 120 minutes of infusion for all sheep”
- Line 130-132: I think it is missing a comma “During and after the infusion, blood”. No need for the “:”.
- Line 133: I think that “samplings” is a more adequate word than “collections”.
- Line 134: “(T8 to T13)”.
- Line 136: Why you indicate here the State, but you did not indicate it in Line 111?
- Line 136: Sao Paulo is not written always in the same way.
- Line 140: missing a stop before “All sheep”
- Line 140: Fully recovered from what?
- Line 142: “from …. to …”
- Line 147: See comments for Line 136.
- Line 145: Check rules around commas and time phrases and correct that trough all the manuscript.
- Line 162-164: Do not use single-sentence paragraph.
- Line 164: To which techniques are referring?

Validity of the findings

Results
- In general, two decimals are enough, report the results between brackets in a way that they can be understood, use point for decimals. For example, lines 198 to 203.
- Line 185-186: Sentence not clear: “non presented any”.
- Line 188-189: Missing space between numbers and units.
- Line 192: maybe better refer to methods instead of tests?
- Line 194: usually 2 decimals are enough, and makes easy to follow.
- Line 194: information between parenthesis difficult to follow, maybe rewrite/reorder.
- Line 194: write the equation to make it easy to follow. For example, add the ×, put a comma to separate the equation from the information between brackets, etc.
- Line 197: Do not start a sentence with an abbreviation.
- Line 197: Check if Bias needs to go with capital letters.
- Line 210: What means 61 points of data?
- Line 212 and Line 215: higher instead of high.
- Line 214: I think it is better “greater” than “further”.
- Line 212-216: Still not clear enough.

Discussion
- Line 219-212: Sentence not clear.
- Line 212: Not clear. “blood concentration was greater than 1.0 mmol/L BHB”?
- Line 225-227: Better to indicate “(61 out of XX; 24.7%)”. Also for the referenced papers in Line 226 and 227.
- Line 231: Revise the sentence.
- Line 234: Remember to use the past tense. (Check all the manuscript for that).
- Line 234-238: Sentence too long
- Line 245: that this was in cattle was already indicated at the beginning of the sentence.
- Line 245-247: Check sentence. Place the comma after BHB.
- Line 247-249: Sentence difficult to follow. Maybe you need a comma in “, and the sensibility”, or you should try to present the same structure in both parts of the sentence “, and high sensibility”.
- Line 250-252: Sentence not clear. To what is referring “the and perfect agreement”? seems something is missing.
- Line 250-253: Avoid creating a single-sentence paragraph.
- Line 261: Missing comma before “the greater”?
- The term “somehow” feels not accurate. I recommend to find another, or at least, do not use it so close in the text. For example, it is in Line 256, and then in Line 262.
- Check the use of the verb “evoke”.
- Line 272-277: Rewrite conclusions. Sentence too long (Line 272-276), and not clear what means the last sentence.

Additional comments

Although the authors have improved the manuscript, there are still some issues they need to improve. I strongly suggest revising carefully the English (in particular the grammar and the way to structure sentences). Moreover, American and British English are mixed in the document. The idea of the material and methods is that other researchers would be able to replicate the study’s conditions, because of that it is important to provide information regarding the composition and chemical analysis of the feed administered.

---

## Round 0.3 · Minor Revisions

Dear authors, please find attached some minor comments that must be addressed according to reviewer 3.

Reviewer 1 ·

Basic reporting

In my opinion the authors made a special effort to respond to all comments, made the corrections and changes suggested by the reviewers in the manuscript.
I wish to congratulate the authors for the work done, for the innovative experimental design and for the results obtained that are reflected in the current manuscript.
Therefore the manuscript meets conditions to be published.

Experimental design

None

Validity of the findings

none

Additional comments

In my opinion the authors made a special effort to respond to all comments, made the corrections and changes suggested by the reviewers in the manuscript.
I wish to congratulate the authors for the work done, for the innovative experimental design and for the results obtained that are reflected in the current manuscript.
Therefore the manuscript meets conditions to be published.

Reviewer 3 ·

Basic reporting

The document has improved. However, some issues are still not clear or need to be revised.
-Check American/British spelling also in tables and figures.
-Reduce decimals in Tables and Figures, and report in the same way the formula (equation) in Tables and in the text.
-Check Tables and Figures for the comma and spot to report decimals.
-Figure 1: Indicate in the title what is the blue line, and the red one. Add in the figure the equation and the r.
-Figure 4: Indicate in the figure when the infusion started and when it finished. Could you report the ES instead of SD, so it is more clear when point differs significantly, or, if you keep using SD indicate which point significantly with the baseline.
-When reporting XX ± XX it is not clear if you are reporting ES or SD. Please make it clear. Moreover, if a P-value is associated with that information it is better to indicate the ES and not the SD.
-Line 39: Here you are reporting a range, not only the threshold. I suggest “BHB concentration range”.
-Line 41: Start a new sentence for severe ketosis.
-Line 83: Correct capital letter used for “It”
-Line 91: sentence not clear, something is lacking.
-Line 110: to what is “(2.7%)” referring?
-Line 117: Could the editor check if this is correct for that journal. Even if using software for adding the references, the journal guidelines (or recently published papers) have to been checked to see how they report citations within the text. Usually “&” is only used inside a parenthesis.
-Line 118: Check the need of adding a comma after “Before the BHB infusion”
-Line 149: there is an extra stop at the end of the sentence
-Line 161-163: A unique sentence cannot be a paragraph. That was not corrected from the previous submission.
-Line 164: Still not clear to which techniques are referring. I suggest stating their name again.
-Line 170: This does not agree with the following sentences. I suggest to delete it.
-Line 192: use the symbol ×, not the x.
-Line 194: showed instead of show.
-Line 197: Not clear “for normal and moderate”.
-Line 227: use “It was” or “they were”.
-Line 232: Use “There are no previous studies”
-Line 242: “this could increase”
-Line 244-245: Difficult to follow.
-Line 246-250: A unique sentence cannot be a paragraph. That was not corrected from the previous submission.

Experimental design

N/A

Validity of the findings

N/A

Additional comments

N/A

---

## Round 0.4 · Minor Revisions

Dear authors,

We would like not to disturb you again, but some minor points are still requested.

Reviewer 3 ·

Basic reporting

Some (very) small comments regarding Table 1:
-Table 1 is indicated in the M&M but it was not at the end of the manuscript with the other Tables.
-Table 1 needs to indicate the units of the parameters (%) and is they are calculated on a DM basis.
-I think that the paper is using British spelling, but in Table 1, “fiber” is written with the American Spelling. In British is “fibre”.

No further comments to add.

Experimental design

No comment

Validity of the findings

No comment

Additional comments

No comment

---

## Round 0.5 · accepted · Accept

Thank you for handling all the comments suggested by reviewers.